# DiffSeg30k: A Multi-Turn Diffusion Editing Benchmark for Localized AIGC Detection

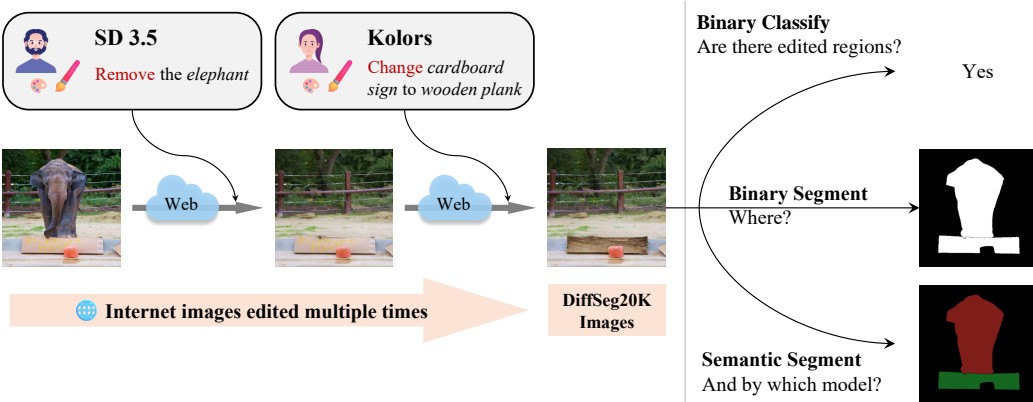

Figure 1: **Motivation.** Images shared online may be independently edited multiple times by different users using different models. DiffSeg30k simulates this real-world scenario through multi-turn diffusion-based edits, enabling novel fine-grained AIGC detection - simultaneous edit localization and model attribution.

## ABSTRACT

Diffusion-based editing enables realistic modification of local image regions, making AI-generated content harder to detect. Existing AIGC detection benchmarks focus on classifying entire images, overlooking the localization of diffusion-based edits. We introduce DiffSeg30k, a publicly available dataset of 30k diffusion-edited images with pixel-level annotations, designed to support fine-grained detection. DiffSeg30k features: 1) In-the-wild images—we collect images or image prompts from COCO to reflect real-world content diversity; 2) Diverse diffusion models—local edits using eight SOTA diffusion models; 3) Multi-turn editing—each image undergoes up to three sequential edits to mimic real-world sequential editing; and 4) Realistic editing scenarios—a vision-language model (VLM)-based pipeline automatically identifies meaningful regions and generates context-aware prompts covering additions, removals, and attribute changes. DiffSeg30k shifts AIGC detection from binary classification to semantic segmentation, enabling simultaneous localization of edits and identification of the editing models. We benchmark two baseline segmentation approaches, revealing significant challenges in segmentation tasks, particularly concerning robustness to image distortions. We believe DiffSeg30k will advance research in fine-grained localization of AI-generated content by demonstrating the promise and limitations of segmentation-based detection methods.

## 1 INTRODUCTION

Detecting AI-generated content (AIGC) holds significant societal importance due to its implications for trust, misinformation, privacy, and copyright protection. With the rapid advancement of generative AI technologies, particularly diffusion-based editing techniques (Bfl.ai, 2024; Stability.ai, 2024;

Table 1: **Comparison with previous AIGC detection benchmarks.** We just compare datasets that includes local editing data.

| Datasets | Image | Availability | # Max Edit Turns | # Total | Diffusion Editing Models Name |
|---|---|---|---|---|---|
| HIFI-Net (Guo et al., 2023) | General | ✓ | 1 | 0 | - |
| COCO Glide (Guillaro et al., 2023) | General | ✓ | 1 | 1 | Glide (Nichol et al., 2021) |
| OnlineDet (Epstein et al., 2023) | General | ✗ | 1 | 3 | SD1 (Rombach et al., 2022), SD2 (Rombach et al., 2022), Adobe Firefly (Adobe, 2024) |
| DA-HFNet (Liu et al., 2024) | General | ✗ | 1 | 2 | InpaintAnything (Yu et al., 2023), Paint by Example (Yang et al., 2023) |
| WFake (Tǎntaru et al., 2024) | Facial | ✓ | 1 | 2 | Repaint-p2 (Lugmayr et al., 2022), Repaint-LDM (Rombach et al., 2022) |
| DiffSeg20k(Ours) | General | ✓ | 3 | 8 | SD2 (Rombach et al., 2022), SD3.5 (Stability.ai, 2024), SDXL (Podell et al., 2023), Flux.1 (Bfl.ai, 2024), Glide (Nichol et al., 2021), Kolors (Team, 2024), HunyuanDiT1.1 (Li et al., 2024), Kandinsky 2.2 (Shakhmatov et al., 2023) |

Podell et al., 2023; Ho et al., 2020), it has become possible to produce highly realistic modifications in localized image regions (Meng et al., 2021; Rombach et al., 2022). Such capabilities raise serious concerns, as these subtle yet realistic edits pose considerable challenges to content verification and digital forensics.

Current methods and benchmarks for AIGC detection predominantly focus on classifying entire images (Bird & Lotfi, 2024; Zhu et al., 2023b; Sha et al., 2023; Verdoliva et al., 2022), with only limited consideration for localized editing scenarios. This has resulted in a notable gap: the lack of systematic benchmarks specifically designed to evaluate precise detection and localization of AI-generated edits, especially those produced using increasingly popular diffusion-based methods (Stability.ai, 2024; Bfl.ai, 2024). Traditional related tasks, such as Image Forgery Localization (IFL) (Guo et al., 2023; Guillaro et al., 2023; Wang et al., 2022; Zhu et al., 2023a; Liu et al., 2022; Yu et al., 2024), typically address edits made using conventional techniques like copy-move or splicing, yet existing IFL benchmarks rarely cover advanced diffusion-based editing approaches.

In this paper, we introduce **DiffSeg30k**, a publicly available dataset comprising 30k diffusion-edited images with pixel-level annotations, specifically aimed at facilitating the detection and localization of diffusion-based edits. To reflect realistic, practical scenarios, DiffSeg30k incorporates four key features: 1) *In-the-wild images*—we collect natural images from COCO dataset (Lin et al., 2014) to ensure real-world content diversity. In addition, we leverage COCO-derived prompts with diffusion models to generate complementary AI-based images; 2) *Diverse diffusion models*—local editing is conducted using eight state-of-the-art diffusion models; 3) *Realistic user-driven edits*—an automated annotation pipeline leveraging vision-language models (VLMs) (Bai et al., 2025) identifies semantically meaningful regions and generates context-aware editing prompts, including object addition, removal, and attribute modification; and 4) *Multi-turn editing*—each image undergoes up to three sequential edits using different diffusion models, simulating realistic scenarios in which online-shared images are iteratively modified (see Figure 1). Beyond pixel-level annotations of edited areas, the specific diffusion model used for each edit is annotated, defining a novel semantic segmentation task for fine-grained localization that simultaneously identifies the edited regions and the corresponding editing models. Table 1 compares DiffSeg30k with several existing benchmarks containing local editing data.

Leveraging DiffSeg30k, we benchmark two baseline segmentation models—FCN (Long et al., 2015) and Deeplabv3+ (Chen et al., 2017; 2018)—on both binary and semantic segmentation tasks. Our preliminary experiments reveal several key findings: (1) representational capacity is critical, as FCN fails on both tasks; (2) semantic segmentation in multi-turn editing scenarios remains highly challenging; (3) baseline models are sensitive to post-hoc image transformations such as resizing and JPEG compression; and (4) Deeplabv3+ exhibits strong generalization to unseen generators, indicating promising potential for building more generalizable detection models. We hope the release of DiffSeg30k and these initial results will draw community attention to the important problem of diffusion-based editing localization and model attribution, while highlighting the challenges ahead.

## 2 RELATED WORK

### 2.1 IMAGE FORGERY LOCALIZATION

This task focuses on detecting image regions modified by traditional editing techniques such as copy-move and splicing. Representative benchmarks include CASIA (Dong et al., 2013), Columbia (Ng et al., 2009), NIST16 (nis, 2016), IMD20 (Novozamsky et al., 2020), Coverage (Wen et al., 2016),

Figure 2: **Automatic data collection framework.** The pipeline consists of two stages: (1) identifying editable regions using VLMs (Bai et al., 2025) and Grounded-SAM (Ren et al., 2024a) to generate object masks; (2) generating context-appropriate editing prompts with VLMs and applying sequential diffusion-based edits to the selected regions.

FantasticReality (Kniaz et al., 2019), PSCC-Net (Liu et al., 2022), OpenForensics (Le et al., 2021), etc. Based on these datasets, various approaches have been developed, leveraging transformer-based feature extraction (Guillaro et al., 2023; Wang et al., 2022), progressive learning (Zhu et al., 2023a; Liu et al., 2022), hierarchical mechanisms (Guo et al., 2023), and diffusion priors (Yu et al., 2024) to localize edited areas.

## 2.2 AIGC Detection

The task of AIGC detection involves identifying content created by generative AI models such as GANs and diffusion models. Current datasets (Zhu et al., 2023b; Sha et al., 2023; Bird & Lotfi, 2024; Verdoliva et al., 2022; Wang et al., 2023; He et al., 2021; Guo et al., 2023) and methods (Cozzolino et al., 2024a; Wang et al., 2020; Zhong et al., 2023; Cozzolino et al., 2024b) predominantly focus on classifying entire images. Existing datasets can be categorized into two types according to their image content: facial images (He et al., 2021; Yang et al., 2019; Wang et al., 2019; Dang et al., 2020; Gandhi & Jain, 2020) and general images (Zhu et al., 2023b; Sha et al., 2023; Bird & Lotfi, 2024; Verdoliva et al., 2022; Wang et al., 2023). Several datasets (Guillaro et al., 2023; Tânțaru et al., 2024; Epstein et al., 2023; Liu et al., 2024) include limited numbers ($\leq 3$) of diffusion models for localized edits, as detailed in Table 1. In contrast, our proposed DiffSeg30k dataset focuses on general images, covers eight different diffusion models, and incorporates multi-turn editing scenarios. This comprehensive dataset provides systematic support for advancing fine-grained diffusion-based localization tasks.

## 3 DiffSeg30k benchmark

### 3.1 Dataset construction

We developed an automated image editing and annotation pipeline based on vision-language models (VLM), illustrated in Figure 2. The pipeline comprises two stages: 1) generating candidate editing regions and 2) creating suitable editing prompts for these regions, followed by image editing.

**Stage 1—generating candidate editing regions.** As shown in Figure 2, given an arbitrary image, we first use Qwen2.5-VL (Bai et al., 2025) to identify distinct objects present in the image. The identified object categories are then provided to Grounded-SAM (Ren et al., 2024a) to generate corresponding masks for each object. Subsequently, we calculate the Intersection-over-Union (IoU) among object masks and eliminate redundant masks with IoU greater than 70% to avoid repetitive editing of the same object. Finally, we randomly select between one to three object masks for editing.

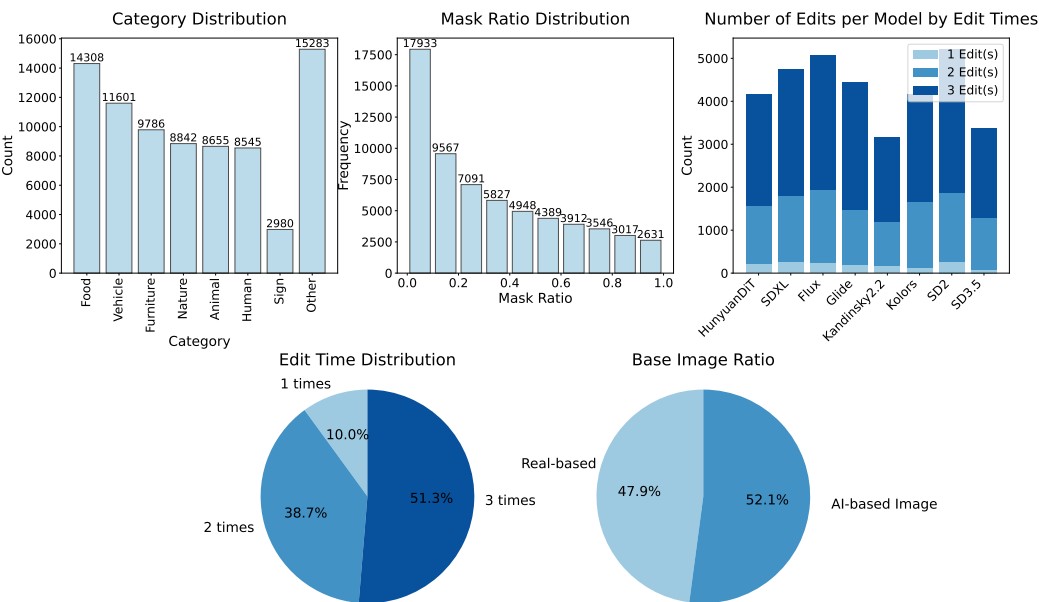

Figure 3: **DiffSeg30k Statistics**. The distributions reflect our balanced data collection strategy: (a) Object category distribution shows Food (14,308) are most frequently edited, with an overall balanced distribution across categories. (b) Mask area ratio distribution reveals a decreasing trend from smaller to larger regions. (c) Editing count per model indicates uniform participation of all eight diffusion models across first, second, and third editing rounds. (d) We manually increased the frequency of multi-turn editing, with the first, second, and third rounds occurring at an approximate 1:4:5 ratio, as multi-turn editing subsumes single-turn cases and presents a more challenging scenario. (e) The base image ratio is approximately balanced between real and AI-generated images.

**Stage 2—editing prompt generation & inpainting.** As depicted in Figure 2, each selected object mask from Stage 1 is paired with the original image and fed into the Qwen2.5-VL model, prompting it to generate contextually appropriate editing prompts. For inpainting, the generated editing prompt, corresponding object mask, and the original image are simultaneously provided to a diffusion model to perform localized inpainting. For multi-turn editing, the edited image can form a new pair with the remaining object masks, repeating the prompt generation and inpainting processes. Thus, we obtain images edited sequentially up to three times, depending on the number of masks selected in Stage 1.

**Data diversity & balance.** To ensure the diversity and balance of the edited images, we specifically considered the following six aspects: 1) *Balanced Candidate Object Types:* We prompted the VLM to prioritize selecting humans, as human-centric edits are significant in practical applications, yet VLMs tend to under-select humans naturally. 2) *Balanced Candidate Mask Area:* We encouraged VLM to prioritize larger objects for editing, counteracting the empirical tendency of VLM to select smaller objects. 3) *Balanced Edit Types:* During editing prompt generation, we prompted VLM to randomly choose from three edit types: attribute changes, object additions, and object removals. For object additions, we followed the approach of SEED-Data-Edit (Ge et al., 2024), first removing an object and then adding a new object in the same position to ensure natural placement. 4) *Balanced Edit Models:* For inpainting, we randomly selected one diffusion model from eight state-of-the-art diffusion models to execute an editing prompt. 5) *Enhanced Multi-turn Editing:* We manually increased the frequency of multi-turn editing, with the first, second, and third turns occurring at an approximate 1:4:5 ratio, as multi-turn editing subsumes single-turn cases and presents a more challenging scenario. 6) *Balanced Real and AI Bases:* Approximately half of the base images were real, collected from COCO, while the other half were AI-generated using COCO prompts with random diffusion models.

## 3.2 SANITY CHECK

Our automated pipeline occasionally produces low-quality edits (see Figure 8) due to (i) limited diffusion editing capability (e.g., incomplete object removal) or (ii) mask errors from Grounded-SAM (Ren et al., 2024a). These issues largely stem from our framework's ambition to simulate diverse and realistic editing scenarios. In contrast, existing datasets such as OnlineDet (Epstein et al., 2023) rely on random masks with potentially empty prompts, while wFake (Țânțaru et al., 2024) focuses only on predefined facial regions (e.g., eyes, nose). Our approach covers a broader and more complex range of edits, though at the cost of occasional artifacts. To improve dataset quality, we employ Qwen2.5-VL (Bai et al., 2025) to assign image-level quality scores (0–5) and automatically discard edits with severe artifacts (score $< 3$). Details of the scoring prompt and filtered images are provided in Appendix B.

## 3.3 DATASET STATISTICS

We conducted a statistical analysis of DiffSeg30k, illustrated in Figure 3. It reveals that smaller objects are edited more frequently, whereas larger objects tend to be edited less often, with editing frequency smoothly decreasing as object area increases. Objects such as food, vehicles, furniture, nature scenery, animals, humans, and signs are among the most frequently edited. The dataset maintains a balanced distribution across editing models and base image distribution. Figure 4 shows some editing results.

# 4 BENCHMARKING BASELINES

## 4.1 EXPERIMENTAL SETUP

**Editing data preparation.** To match the input requirements of diffusion editing models as well as achieving high editing quality, COCO images are first resized so that the shorter side is 1024 pixels, followed by a center crop to 1024×1024. For model-specific compatibility, images are resized to 256 for Glide during editing.

**Training details.** Due to the lack of existing baseline segmentation models specifically for general AIGC localization, we trained two baselines based on classic semantic segmentation frameworks: FCN-8s (Long et al., 2015) and Deeplabv3+(Chen et al., 2017; 2018) with ResNet50(He et al., 2016). We evaluated these models on two tasks: binary segmentation (localizing edited areas) and semantic segmentation (localizing edited areas while distinguishing among different editing models). The dataset is split into training and validation sets in an 8:2 ratio, and all results are reported on the validation set. Models are trained on 512×512 resolution (for acceleration) using default hyperparameters from the following open-source implementations.[1][2] We additionally include 5k unedited real COCO images in the training set to help suppress false positives on normal images. Training FCN-8s and Deeplabv3+ takes approximately 8 and 3 hours, respectively, on an NVIDIA 4090 GPU.

**Evaluation metric.** To evaluate classification performance, we report accuracy (Acc) and mean average precision (mAP). For segmentation, we report pixel-wise accuracy (Acc), mask Intersection-over-Union (mIoU) and boundary-F1 score (bF1) between predicted and ground truth masks. The background class is excluded from mIoU and bF1 calculations to specifically reflect models' abilities to detect edited areas.

## 4.2 RESULTS

**Model performance diverges.** As shown in Table 2, FCN-8s struggles with both binary and semantic localization of diffusion-edited regions. In contrast, Deeplabv3+, a more powerful segmentation model, performs substantially better, achieving strong binary localization with an mIoU of 0.974. These results highlight the importance of model representational capacity.

---

[1] https://github.com/wkentaro/pytorch-fcn
[2] https://github.com/VainF/DeepLabV3Plus-Pytorch

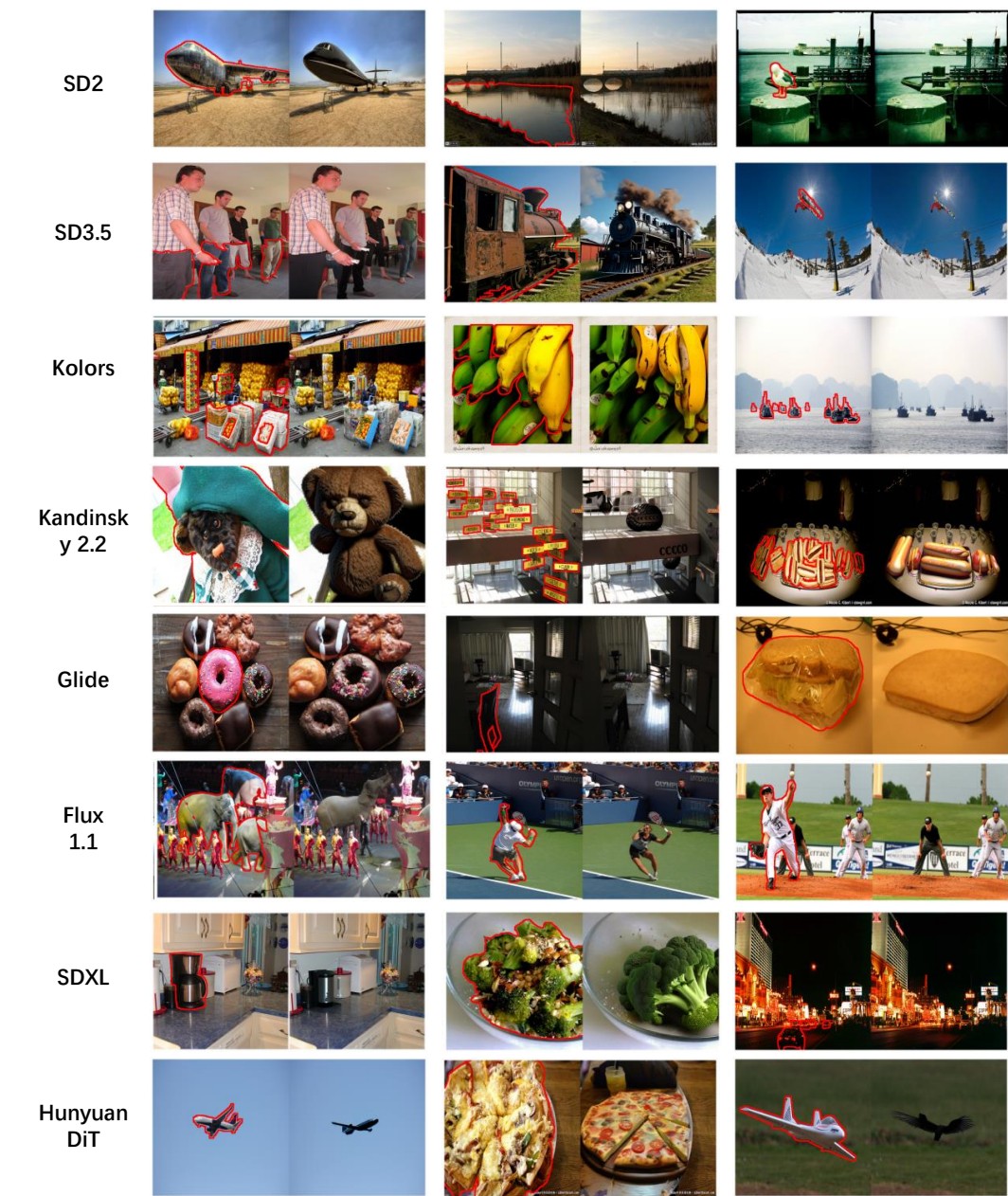

Figure 4: **Example edited images.** For each image pair, the left shows the original image with red contours marking the regions to be edited, and the right presents the corresponding editing result.

**Semantic segmentation is challenging.** Simultaneously localizing edited regions and attributing them to the correct diffusion model is substantially more difficult than binary localization. Even for Deeplab v3+, the mIoU for semantic segmentation drops sharply to 0.76, leaving significant room for improvement. Figure 5 shows the confusion matrices: FCN-8s tends to misclassify edits from all models as background pixels, while Deeplab v3+ struggles particularly with Kolors and SDXL.

Figure 6 further illustrates representative results. FCN-8s produces largely random outputs in both tasks, whereas Deeplab v3+ achieves more coherent segmentations but occasionally misclassifies the editing model.

Table 2: **Performance of baseline segmentation models.** Results of FCN-8s and Deeplabv3+ on binary segmentation (localizing edited regions) and semantic segmentation (localizing and attributing edits to specific diffusion models).

|  | Binary segmentation | | | Semantic segmentation | | |
| --- | --- | --- | --- | --- | --- | --- |
|  | Acc | mIoU | bF1 | Acc | mIoU | bF1 |
| FCN-8s (Long et al., 2015) | 0.566 | 0.366 | 0.192 | 0.363 | 0.053 | 0.039 |
| Deeplab v3+ (Chen et al., 2017) | 0.984 | 0.974 | 0.761 | 0.916 | 0.760 | 0.431 |

Table 3: **Robustness to image transformations.** We evaluate DeepLabv3+ under JPEG compression with varying quality factors and across different input resolutions to assess their sensitivity to common post-processing operations. Baseline indicates no distortions.

|  | | Binary segmentation | | | Semantic segmentation | | |
| --- | --- | --- | --- | --- | --- | --- | --- |
|  | Distortion | Acc | mIoU | bF1 | Acc | mIoU | bF1 |
| Deeplab v3+ (Chen et al., 2017) | Baseline | 0.984 | 0.974 | 0.761 | 0.916 | 0.760 | 0.431 |
|  | JPEG 60 | 0.867 | 0.782 | 0.556 | 0.184 | 0.013 | 0.019 |
|  | JPEG 80 | 0.907 | 0.847 | 0.613 | 0.190 | 0.014 | 0.019 |
|  | resize 256 | 0.811 | 0.751 | 0.631 | 0.021 | 0.012 | 0.021 |
|  | resize 1024 | 0.816 | 0.729 | 0.499 | 0.168 | 0.168 | 0.014 |

**Baseline models are sensitive to image transformations.** We evaluated the robustness of DeepLabv3+ under common image transformations, including JPEG compression and resizing. Specifically, we applied additional transformations to the validation set: JPEG compression with quality factors of 60 and 80, and resizing to 256 and 1024 pixels (models were trained at 512). As shown in Table 3, both JPEG compression and resizing substantially degraded the performance of Deeplabv3+, with severe drops in mIoU—semantic segmentation nearly failed entirely. This underscores the need for improved network designs or data augmentation strategies to develop deployable localization models.

**Do LoRAs affect localization?** Diffusion models can be enhanced with Low-Rank Adaptation (LoRA) modules (Hu et al., 2022) to improve efficiency or expressiveness. In this experiment, we investigate whether the use of LoRAs impacts the localization performance of baseline segmentation models. Specifically, we apply the Hyper-SD LoRA (Ren et al., 2024b) (8 steps, DDIM scheduler (Song et al., 2020)) to SDXL (Podell et al., 2023), which reduces the number of inference steps from 50 to 8. We re-edit all regions in the validation set that were previously modified by SDXL using the LoRA-accelerated version (SDXL-LoRA). Then, we re-evaluate the baseline semantic segmentation model (trained on 8 vanilla generators from the main text, which just saw SDXL-edited images during training). In this setup, edits made by SDXL-LoRA are considered correctly predicted if classified as SDXL, allowing us to evaluate the segmentation model's generalization ability to LoRA-based variants. As shown in Table 4, the segmentation model exhibits strong generalization to LoRA variants, with only a minor mIoU drop of approximately 0.03.

Table 4: **Effect of LoRA on localization performance.** We evaluate semantic segmentation model Deeplabv3+, trained on SDXL-edited images and validated on SDXL- and SDXL-LoRA-edited images. We report mIoU only on masks from SDXL or SDXL-LoRA edits to isolate LoRA's impact. Gray colors the baseline.

|  | Edit model (in val) | mIoU (SDXL) |
| --- | --- | --- |
| Deeplab v3+ (Chen et al., 2017) | SDXL | 0.657 |
|  | SDXL-LoRA | 0.630 |

**Deeplabv3+ achieves strong cross-generator generalization.** Table 5 presents the cross-generator generalization results of Deeplabv3+. Training was conducted on images edited by six diffusion models and tested on edits from two unseen models, with three different combinations evaluated. Key findings are as follows: (1) Deeplabv3+ demonstrates strong cross-generator generalization, with mIoU

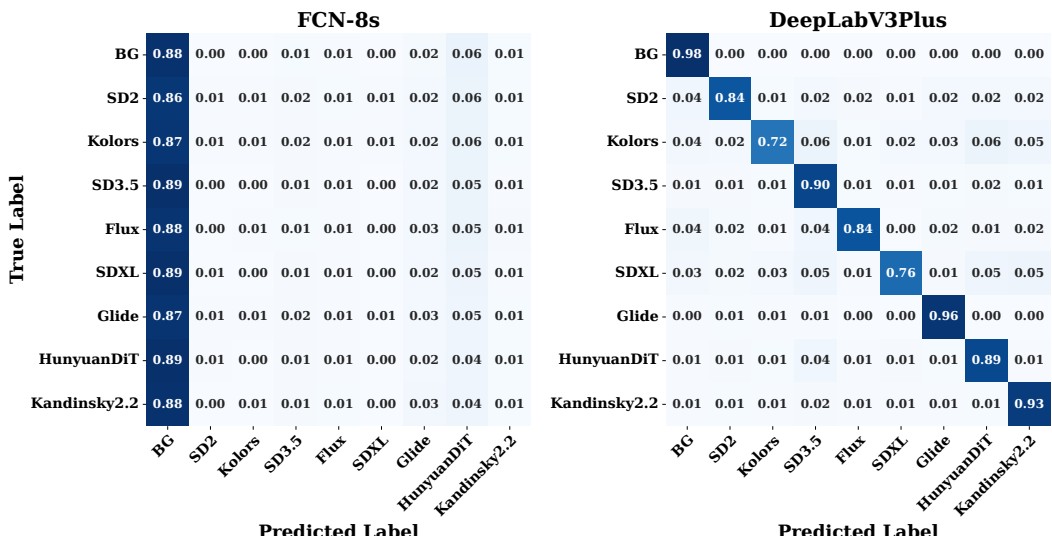

Figure 5: **Confusion matrix on the semantic segmentation task.**

Table 5: **Generalization to unseen diffusion generators**. Segmentation models are trained on images edited by 6 diffusion generators and evaluated on edits from 2 unseen generators to assess cross-generator generalization. Baseline indicates training and testing on 8 generators.

| | | Binary segmentation | | |
|---|---|---|---|---|
| | Generalize to | Acc | mIoU | bF1 |
| | Baseline | 0.984 | 0.974 | 0.761 |
| Deeplab v3+ (Chen et al., 2017) | Flux_Hunyuan | 0.920 | 0.867 | 0.526 |
| | SD2_SD3.5 | 0.931 | 0.833 | 0.372 |
| | Glide_Kolors | 0.953 | 0.927 | 0.648 |

consistently above 0.8; for the Glide-Kolors combination, the mIoU even reaches 0.927. Notably, Deeplab v3+ was not specifically designed for such generalization, suggesting that segmentation-based methods hold great potential for transferable AIGC detection. (2) The combination of SD2 and SD3.5 proved the most challenging, yielding the lowest mIoU.

**Performance of existing whole-image classification models.** Given the abundance of established methods for classifying whole AI-generated images, we further evaluated their effectiveness on our DiffSeg30k, where models must answer a more challenging question — whether there are diffusion edited regions in an image instead of whether an entire image is generated by a diffusion model. We benchmarked two classic classification models: CNNSpot (Wang et al., 2020) and UniversalFakeDet (Ojha et al., 2023). During training, we collect another 30k real images from COCO (Lin et al., 2014) as negative examples. Results in Table 6 show that CNNSpot performs well, while UniversalFakeDet has high false-positive rates, indicating substantial room for improvement. Notably, although UniversalFakeDet was originally designed for cross-model generalization in whole-image classification, it also demonstrates good generalization when detecting local edits, as evidenced by the results in Table 7.

Table 6: **Binary classification results of established AIGC detectors.** Models are evaluated on the task of identifying whether an image contains any local region edited by diffusion models.

| | Acc | mAP |
|---|---|---|
| CNNSpot (Wang et al., 2020) | 0.942 | 0.979 |
| UniversalFakeDet (Ojha et al., 2023) | 0.860 | 0.934 |

Table 7: **Cross-generator generalization of UniversalFakeDet.** The model is trained on images edited by 6 diffusion generators and evaluated on images edited by 2 unseen diffusion generators to assess its ability to generalize across different editing sources. Baseline indicates training and testing on 8 generators.

|  | Generalize to | Acc | mAP |
|---|---|---|---|
| UniFakeDet (Ojha et al., 2023) | Baseline | 0.860 | 0.934 |
|  | Flux_Hunyuan | 0.768 | 0.877 |
|  | SD2_SD3.5 | 0.714 | 0.817 |
|  | Glide_Kolors | 0.749 | 0.844 |

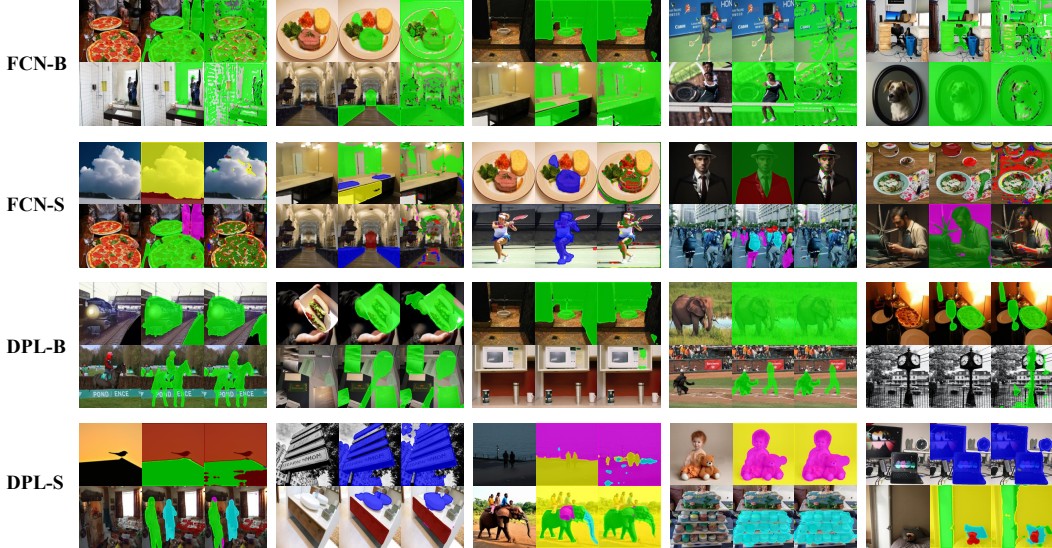

Figure 6: **Segmentation results.** "DPL" denotes Deeplab v3+. "-B" and "-S" refer to binary and semantic segmentation, respectively. In each image group, the first column shows edited images, the second column shows the ground truth masks, and the third column shows the predicted masks.

## 5 CONCLUSION

We introduce DiffSeg30k, a multi-turn diffusion editing dataset that provides a robust foundation for systematically studying diffusion-based editing localization and model attribution. Through extensive experimental analyses, we highlight both the strengths and current limitations of segmentation methods, laying the groundwork for further research in robust AIGC localization techniques.

**Broader Impact.** This benchmark also contributes positively to digital forensics, misinformation mitigation, and media authenticity verification. Nonetheless, we recognize potential risks associated with releasing a comprehensive dataset of edited images, as it may inadvertently reveal detection vulnerabilities to adversaries. To address this concern, our dataset prioritizes transparency and aims to foster the advancement of stronger detection frameworks rather than enabling evasion strategies.

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

## A  LLM USAGE

The authors used large language model as a general-purpose assist tool to aid in polishing the writing of this paper. The LLM was employed to: 1) Refine grammar, phrasing, and readability of draft sections. 2) Improve clarity, coherence, and conciseness without altering the underlying technical content. 3) Suggest alternative wordings to meet the style and professionalism standards of a top AI conference. The LLM was not involved in research ideation, experimental design, analysis, or the generation of scientific claims. All technical content, experimental results, and conclusions are solely the work of the authors.

## B  QUALITY ASSESSMENT

To remove severely low-quality outputs generated by the automated editing framework, we employ Qwen2.5-VL (Bai et al., 2025) to evaluate the quality of each edited image. Figure 7 shows two chain-of-thought demonstration examples provided to Qwen2.5-VL. Images receiving a score lower than 3 are discarded, resulting in the removal of roughly 50% of the generated samples.

**COT 1**

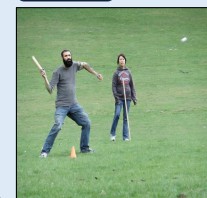

**User:** Are there unnatural things in this image and rate it accordingly from 0 (very unnatural) to 5 (natural and high quality). Please respond with: "X(score), the reason is that [brief explanation in one sentence]."
**Assistant:** 4, the reason is that the man's pose is slightly unnatural but overall scene appears reasonable and natural.

**COT 2**

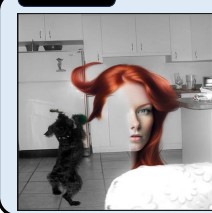

**User:** Are there unnatural things in this image and rate it accordingly from 0 (very unnatural) to 5 (natural and high quality). Please respond with: "X(score), the reason is that [brief explanation in one sentence]."
**Assistant:** 1, the reason is that the red-haired woman's face completely mismatches the context and appears as an incomplete fragment, making the image very unnatural.

**Query**

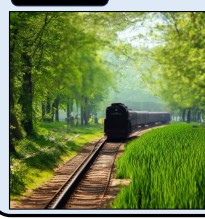

**User:** Are there unnatural things in this image and rate it accordingly from 0 (very unnatural) to 5 (natural and high quality). Please respond with: "X(score), the reason is that [brief explanation in one sentence]."
**Assistant:** XXX

Figure 7: **Chain-of-thought examples for image quality assessment.**

Figure 8 shows typical low-quality edits, usually caused by insufficient diffusion editing capability or inaccurate masks. In the third example, we observe that Qwen2.5-VL occasionally assigns quality scores based on the entire image rather than the edited region, which may inadvertently filter out some higher-quality edits. However, this effect is minor in our dataset construction, as the majority of low-quality edits are successfully removed.

Figure. 9, 10 present additional visualizations of the final edited images after quality filtering from the DiffSeg30k dataset, showcasing the diversity of edit types and visual characteristics across different diffusion models.

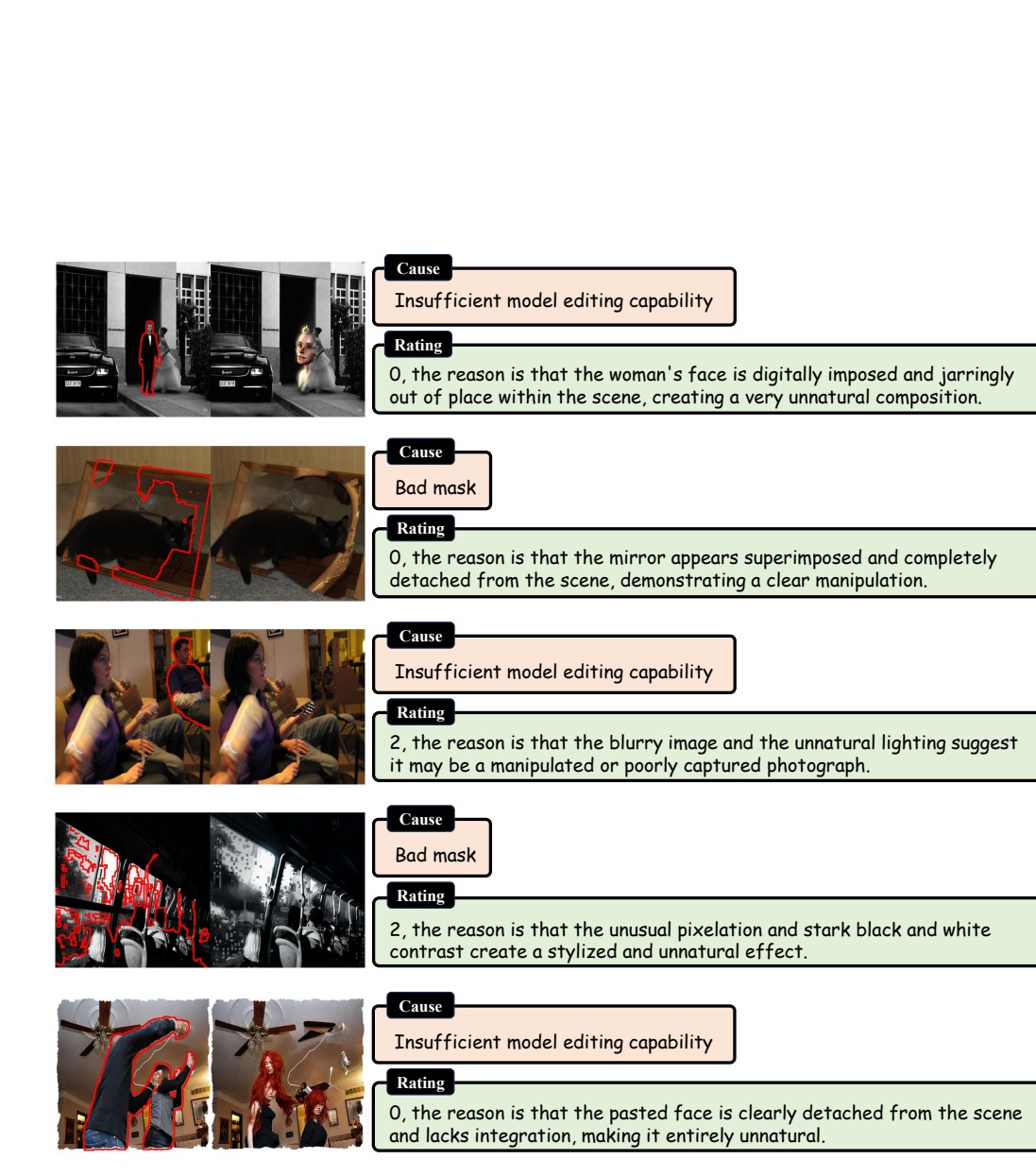

Figure 8: **Typical low quality edits and VLM ratings.**

Glide

Kolors

SD 3.5

Flux.1

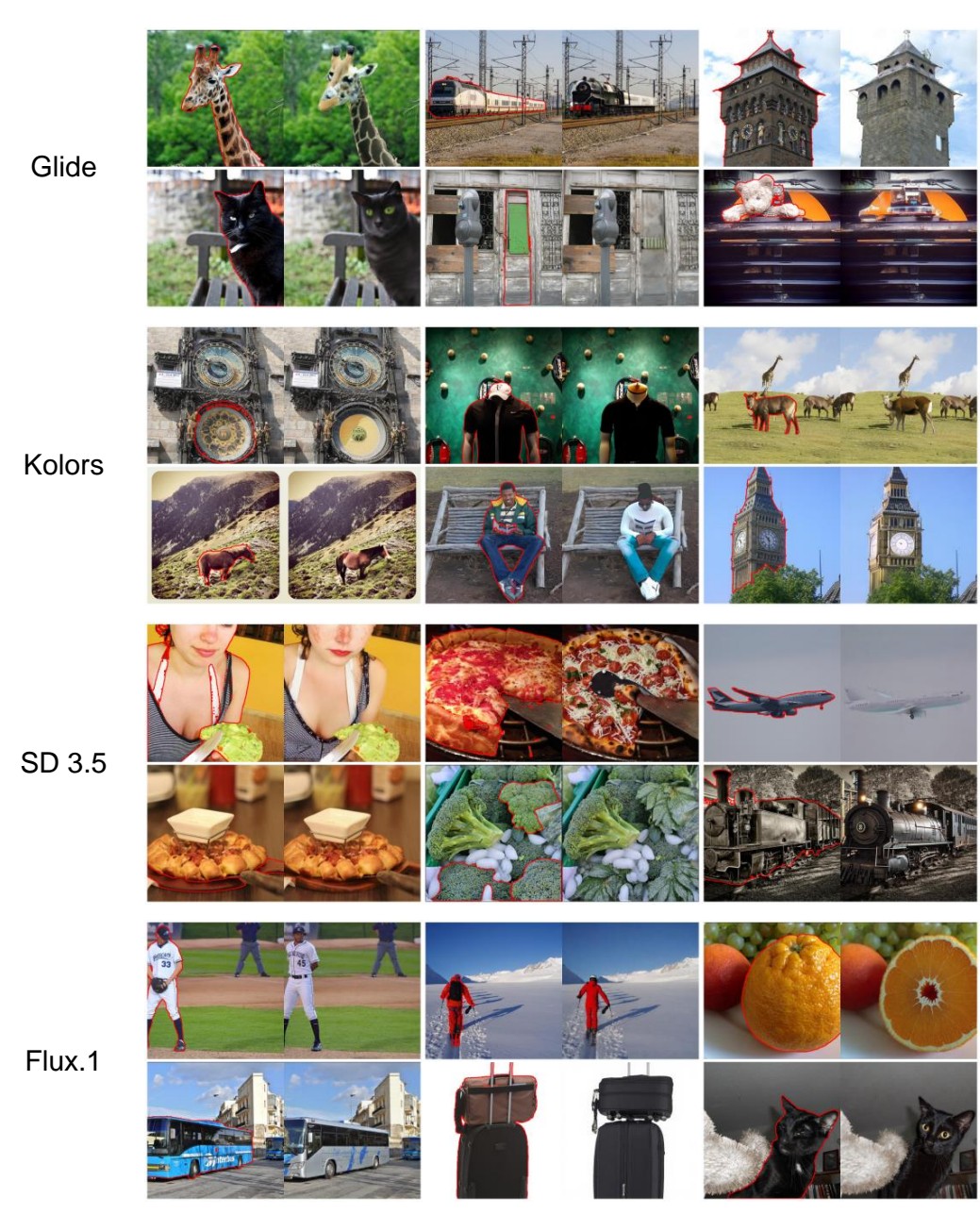

Figure 9: **More examples in DiffSeg30k.** For each pair of images, the left is the original image with red contours highlighting areas to be edited. The right is the editing results.

SD 2

Hunyuan DiT

Kandin-sky 2.2

SDXL

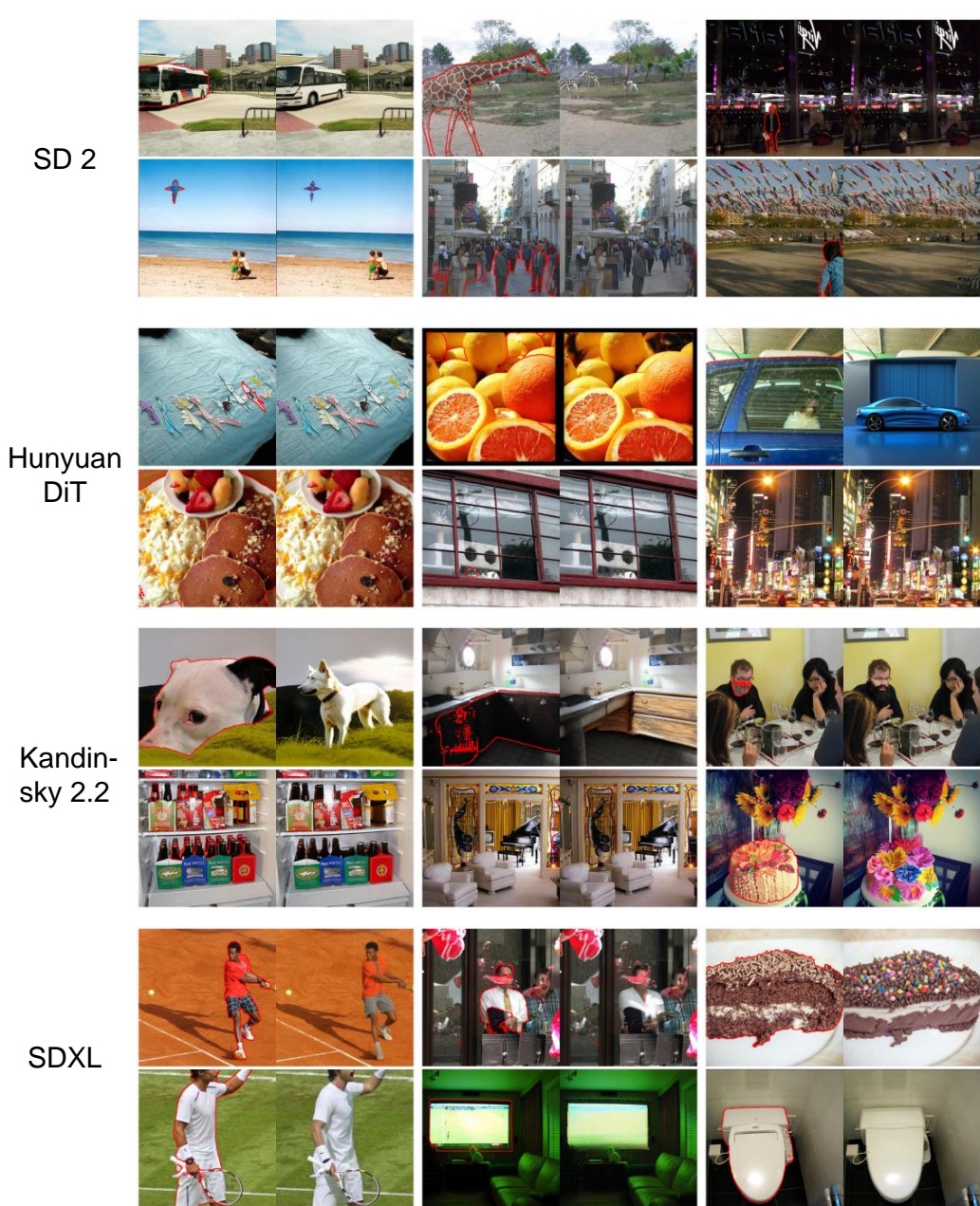

Figure 10: **More examples in DiffSeg30k.** For each pair of images, the left is the original image with red contours highlighting areas to be edited. The right is the editing results.