# OpenReview forum: "DiffSeg30k: A Multi-Turn Diffusion Editing Benchmark for Localized AIGC Detection"
_ICLR.cc/2026/Conference — ICLR 2026 Conference Withdrawn Submission_

### Official Review · Reviewer_TTCL · 2025-10-17

**Soundness:** 3
**Presentation:** 3
**Contribution:** 3
**Rating:** 4
**Confidence:** 3

**Summary:**

This paper presents DiffSeg30k, a new benchmark for localized detection of diffusion-based image edits. Unlike prior datasets focusing on whole-image classification, DiffSeg30k enables pixel-level localization and model attribution. It contains 30,000 images with multi-turn edits (up to three rounds) using eight diffusion models. The authors propose an automated pipeline using Qwen2.5-VL and Grounded-SAM to identify editable regions and generate context-aware edit prompts. Two segmentation baselines (FCN-8s, Deeplabv3+) are evaluated, showing that binary localization is feasible but semantic attribution remains challenging.

**Strengths:**

- Comprehensive benchmark addressing a key unmet need in AIGC detection
- Automated, reproducible pipeline combining VLMs and diffusion models
- Multi-turn editing and diverse diffusion sources add realism
- Transparent evaluation of baseline performance and robustness

**Weaknesses:**

- The proposed benchmark focuses mainly on localized edits and does not effectively address global or stylistic transformations, which are also important for comprehensive AIGC detection.
- While object addition and removal edits are valid operations, they may be less representative of subtle AIGC manipulations compared to attribute-change edits that alter visual details without structural changes.
- Despite the automated quality filtering process, some residual artifacts or dataset biases may persist and could influence model training or evaluation outcomes.
- It would be helpful if the authors clearly state the dataset licensing terms and distribution policy, especially given that DiffSeg30k combines COCO-based real images and diffusion-generated data. Transparent licensing will be important for downstream research use and ethical data sharing.

**Questions:**

1. How does the benchmark handle global or style-based edits that lack localized masks? Would such cases be misclassified as unedited?
2. Object addition and removal cases are valid editing types, but they may not reflect subtle AIGC manipulations. How do you view their relevance compared to attribute-change edits for advancing AIGC understanding?
3. Does the dataset include metadata such as edit order, model parameters, or prompts to support causal or interpretability analyses?

4. [MINOR - discussion] Could incorporating diffusion-aware models provide stronger baselines than standard segmentation networks?

Typo:
- Tab. 1 last row -> "DiffSeg20k"

---

### Official Review · Reviewer_iGB6 · 2025-10-26

**Soundness:** 2
**Presentation:** 3
**Contribution:** 2
**Rating:** 2
**Confidence:** 5

**Summary:**

This paper introduces DiffSeg30K, a benchmark designed for localized detection and attribution of diffusion-based AIGC. DiffSeg30K targets pixel-level edit localization and diffusion model identification under realistic multi-turn editing scenarios. The dataset contains 30K images with pixel-wise annotations edited by eight diffusion models, each potentially edited up to three times. Experiments using FCN and Deeplabv3+ demonstrate that while binary localization is feasible, semantic segmentation remains highly challenging, especially under distortions and multi-model scenarios.

**Strengths:**

1. The paper defines a practically relevant and comprehensive task, covering AIGC detection, localization, and model attribution.
2. The dataset design is systematic, simulating multi-turn edits across eight diffusion models via an automated pipeline.

**Weaknesses:**

1. The dataset is not the first to address AIGC detection, localization, and attribution. The paper should more carefully discuss or experimentally compare against existing datasets, clarifying its unique contributions.
2. The automatic annotation pipeline may introduce noise and bias. Although low-quality samples are filtered using Qwen2.5-VL, the paper does not quantify annotation accuracy. The fact that roughly 50% of samples were discarded raises concerns about the stability of the generation process. Does this benchmark require some level of human validation to ensure quality?
3. The evaluation scope is narrow, and only two segmentation baselines (FCN, Deeplabv3+) are tested. The study omits comparisons with stronger architectures and does not evaluate existing AIGC detection methods that already target detection, localization, and attribution.
4. The significant performance drop in segmentation lacks analysis of potential causes.

**Questions:**

1. Issues raised in the *Weaknesses* section.
2. Was human validation performed during dataset construction or quality control?
3. Have the authors considered evaluating stronger segmentation architectures or specialized AIGC detection models?
4. What performance improvements were observed in existing AIGC detection methods by using the proposed dataset?

---

### Official Review · Reviewer_uL3Z · 2025-10-28

**Soundness:** 2
**Presentation:** 3
**Contribution:** 2
**Rating:** 6
**Confidence:** 2

**Summary:**

The paper moves beyond AI-generated vs. real images classification tasks, and turns to consider fine-grained localization and model attribution of diffusion-based image edits for an arbitrary image. Specifically, the paper introduces a benchmark of 30k images named DiffSeg30K with pixel-level annotations for multi-turn diffusion editing. Each base image (could be real or AI-generated) is edited sequentially by 1 of 8 off-the-shelf diffusion models (SD series, Flux, Glide, Kolors etc) with up to 3 turns. A vision-language pipeline (Qwen 2.5-VL + Grounded-SAM) automatically chooses editable regions and generates semantically meaningful prompts (add/remove/change). Low-quality results are filtered using a VLM-based quality score. The paper also present baseline fine-grained detection results based on FCN-8s, deeplab-v3 and existing ai detectors models.

**Strengths:**

+ The paper proposes a new (to my knowledge) perspective that considers fine-grained localization and model attribution of diffusion-based image edits for an arbitrary image, instead of doing binary classification between real and AI-generated images.
+ The paper proposes a corresponding automatic pipeline for collecting annotated images and present a 30k dataset suitable for the proposed tasks.
+ The evaluation looks abundant on FCN-8s and deeplab-v3 with additional results on binary ai content classifier.
+ The paper is generally well-written and easy to understand.

**Weaknesses:**

- Although the paper proposes a new point of view moving from binary detection to fine-grained model attribution, can the author comment on how this new task could concretely further benefit the research community?
- The current baseline results focus primarily on fcn-8s and deeplab-v3 models. We can observe significant improvements in terms of detection rates with better architectures. Therefore what if we use more advanced architectures and models such as ViT based detectors or other segmentation models that prove better empirical results? Will these better detectors saturate on the collected data?
- Admittely, I am not very familiar with the current progress of this particular sub-domain. I would therefore also like to hear other colleague reviewers' opinions.

**Questions:**

Please refer to the weaknesses section for detailed questions. Thanks

---

### Official Review · Reviewer_fcVC · 2025-10-29

**Soundness:** 2
**Presentation:** 2
**Contribution:** 1
**Rating:** 2
**Confidence:** 4

**Summary:**

For AI-Generated Content (AIGC) detection and localization, the proposed DiffSeg30K targets determining whether an image has been edited by generative models and segmenting the edited pixels under realistic multi-turn workflows. The dataset is constructed via a two-stage automatic pipeline: (i) a Vision–Language Model (VLM) with Grounded Segment Anything Model (Grounded-SAM) identifies editable regions and produces corresponding tags/masks; (ii) using these tags/masks as context, the VLM generates randomized editing instructions that diverse text-to-image diffusion models execute over multiple rounds to synthesize edited images, thereby producing multi-turn image/mask pairs tailored for AIGC.

**Strengths:**

* The paper constructs a dataset that moves beyond single-shot edits to support multi-turn editing workflows.
* DiffSeg30K provides an easily reproducible pipeline using only pre-trained models such as VLM (e.g., Qwen2.5-VL), open-vocabulary segmentation (e.g., Grounded-SAM), and text-to-image diffusion models (e.g., SDXL) without requiring additional training.

**Weaknesses:**

**1. Limited novelty in dataset pipeline.** The current pipeline relies on vision-language models (e.g., Qwen2.5-VL) and open-vocabulary segmentation (e.g., Grounded-SAM) to generate pseudo labels before multi-turn editing. Similar pseudo-labeling strategies that couple region mining for text-to-image diffusion models are already common in existing studies [1, 2]. For technical contribution, Please consider a different labeling mechanism or provide sufficient analysis strongly related to AIGC tasks.

**2. Lack of analyzing the annotation quality of DiffSeg30K.** The dataset is created by a fully automatic pipeline that selects editable regions and masks, generates editing instructions, and applies image edits across multiple rounds. The paper treats the resulting labels as reliable without evidence. There is no quantitative check of mask accuracy from Grounded-SAM, no metric that verifies whether VLM-generated instructions are grounded in the input image, and no measurement of background preservation or edit success. Well-known risks are left unmanaged, including VLM hallucination [3], failures of background preservation for editing [4], unintended object re generation during inpainting [5], and a significant performance gap between seen and unseen classes in open-vocabulary detection models [6]. Errors introduced in the first round can propagate and amplify across later rounds, which directly undermines label trust. At minimum, please define metrics for edit success or use preference-trained classifiers [7] to filter low-quality editing samples. Without these checks, annotation quality remains unverified and the downstream claims are uncertain.

**3. Unclear value of multi-turn compared to single-turn.** The paper asserts the importance of multi-turn editing but does not show benefits beyond what single-turn benchmarks already provide. There is no controlled study that varies only the number of rounds while keeping the edit-type distribution comparable. No application is presented that truly requires a multi-turn edit history rather than a single-turn edit. A detailed comparison or a concrete application demonstrating unique value is needed to justify the added complexity for multi-turn editing benchmarks.

**4. Outdated segmentation baselines.** The paper omits modern segmentation backbones [8, 9] that are standard in benchmark work. Strong ViT-based models (e.g., EOMT [8]) and popular decoders (e.g., Mask2Former [9]) should be included in the Experiment section.

**5. Robustness limited to simple image transforms.** In Table 3, the robustness study only considers JPEG compression and resizing. Following the common protocol [10] for robustness, adversarial corruptions should be part of the evaluation.

**6. Only LoRA analysis without full fine-tuning.** Table 4 reports segmentation shifts after LoRA fine-tuning on SDXL, which is too narrow to claim stability. Full fine-tuning with the same architecture on large-scale data and comparisons across additional generator backbones are needed to assess localization and attribution stability.

**7. Readability.** For example, Figure 1 labels the dataset as “DiffSeg20K” while the title and abstract use “DiffSeg30K”; the naming should be unified. For qualitative examples, pair each panel with the exact instruction and include a concise color legend (see Figure 6). These fixes reduce confusion and make the visual evidence support the claims more clearly.


[1] TokenCompose: Text-to-Image Diffusion with Token-level Supervision, NeurIPS 2024.

[2] CoMat: Aligning Text-to-Image Diffusion Model with Image-to-Text Concept Matching, NeurIPS 2024.

[3] Mitigating Object Hallucination in Large Vision-Language Models via Image-Grounded Guidance, ICML 2025.

[4] Early Timestep Zero-Shot Candidate Selection for Instruction-Guided Image Editing, ICCV 2025.

[5] Attentive-Eraser: Unleashing Diffusion Model's Object Removal Potential via Self-Attention Redirection Guidance, AAAI 2025.

[6] Grounding DINO: Marrying DINO with Grounded Pre-Training for Open-Set Object Detection, ECCV 2024.

[7] Pick-a-Pic: An Open Dataset of User Preferences for Text-to-Image Generation, NeurIPS 2023.

[8] Your ViT is Secretly an Image Segmentation Model, CVPR 2025.

[9] Masked-attention Mask Transformer for Universal Image Segmentation, CVPR 2022.

[10] Anyattack: Towards Large-scale Self-supervised Adversarial Attacks on Vision-language Models, CVPR 2025.

**Questions:**

Q1. What component or analysis is new beyond open-vocabulary pseudo-labeling pipelines? Can you show a head-to-head against alternatives inspired by [1, 2] to justify design choices?

Q2. Will you add a small human feedback (e.g., 300–500 samples) to quantify mask accuracy, instruction grounding, and edit success?

Q3. What concrete metrics will you report for instruction grounding and for edit success/background preservation?

Q4. Under matched edit budgets, does increasing the number of rounds change difficulty or failure modes versus single-turn? Please provide a controlled comparison.

Q5. Will you include modern segmentation backbones and decoders [8, 9]?

Q6. Beyond SDXL with LoRA fine-tuning, please evaluate full fine-tuning with the same architecture and other text-to-image diffusion models for localization/attribution stability.

**Details Of Ethics Concerns:**

The pipeline can generate NSFW or otherwise unsafe content during VLM instruction generation and diffusion inpainting, including sexual or violent edits. Please harden prompting (safety-aware instruction templates and blocklists) and add post-generation filters for edited images (e.g., NSFW/violence classifiers),.

---

### Note · Authors · 2025-11-13

I have read and agree with the venue's withdrawal policy on behalf of myself and my co-authors.